# Immediate versus Delayed Attachment Incorporation Impact on Prosthetic Aftercare among Mandibular Implant—Supported Overdenture Wearers

**DOI:** 10.3390/jcm11123524

**Published:** 2022-06-19

**Authors:** Eran Zenziper, Ofir Rosner, Oded Ghelfan, Joseph Nissan, Sigalit Blumer, Gil Ben-Izhack, Moshe Davidovich, Liat Chaushu, Adrian Kahn, Sarit Naishlos

**Affiliations:** 1Department of Prosthodontics, The Maurice and Gabriela Goldschleger School of Dental Medicine, Tel-Aviv University, Ramat-Aviv, Tel Aviv 69978, Israel; rosnerop@yahoo.com (O.R.); drghelfanoded@gmail.com (O.G.); nissandr@gmail.com (J.N.); gil.ben.izhack@gmail.com (G.B.-I.); 2Department of Pediatric Dentistry, The Maurice and Gabriela Goldschleger School of Dental Medicine, Tel-Aviv University, Ramat-Aviv, Tel Aviv 69978, Israel; blumer@012.net.il (S.B.); river554@gmail.com (S.N.); 3Department of Orthodontics, The Maurice and Gabriela Goldschleger School of Dental Medicine, Tel-Aviv University, Ramat-Aviv, Tel Aviv 69978, Israel; davidom@post.tau.ac.il; 4Department of Periodontics and Implant Dentistry, The Maurice and Gabriela Goldschleger School of Dental Medicine, Tel-Aviv University, Ramat-Aviv, Tel Aviv 69978, Israel; liat.natanel@gmail.com; 5Department of Oral & Maxillofacial, The Maurice and Gabriela Goldschleger School of Dental Medicine, Tel-Aviv University, Ramat-Aviv, Tel Aviv 69978, Israel; dr.adykahan@gmail.com

**Keywords:** mandibular implant overdenture, denture settling, aftercare, immediate activation, delayed activation

## Abstract

Background: Substantial effort is dedicated to finding the most favorable parameters that will ensure low aftercare demands among edentulous patients wearing mandibular implant supported overdentures (MISODs). The purpose of this retrospective cohort study was to compare prosthetic aftercare between MISOD patients with a simultaneous (group A) vs. a three-week settling in period (group B) prior to attachment incorporation. Methods: Forty-five patients enrolled in this study. Two implants per patient were placed using a two-stage implant insertion protocol. Second-stage surgery was performed after three months. All patients received ball attachments using the direct (chairside) incorporation method. Twenty-two patients received their dentures with simultaneous attachment activation and the rest—twenty-three patients—after a three-week settling in period. Patients’ files were scanned for aftercare visits. Outcome parameters included sore spot relief, attachment incorporation, and denture repair. Additionally, gingival index measurements were compared. Confounding factors included age, gender, and implant dimensions. Results: The mean follow-up for the entire cohort was 84 ± 21 months, and the range 39–120 months. The mean number of visits for group A vs. B respectively: pressure sores relieve (3.63 ± 0.84 vs. 3.71 ± 0.61, *p* = 0.581), liner exchange due to loss of retention (2.09 ± 1.03 vs. 2.31 ± 1.04 *p* = 0.487), and gingival index (1.3 ± 0.3 vs. 1.03 ± 0.2, *p* = 0.653) exhibited no statistically significant differences between the tested groups. No statistically significant differences between the groups were also noted for the denture repair aftercare treatments (*p* = 0.318) and the independent variables including age, gender, and implant length. Conclusions: Prosthetic aftercare in MISOD wearers is similar whether a simultaneous or a three-week settling in period for attachment incorporation is applied.

## 1. Introduction

Complete edentulism continues to set a challenge in modern dentistry striving to restore form and function [1,2]. The accepted standard of care calls for rehabilitation of the edentulous mandible with implant-supported overdenture gaining its retention and support from a minimum of two implants [3]. A lot of effort has been dedicated in searching the variables influencing the aftercare for those patients and by that improving their quality of life [4]. These include implants number, attachments incorporating method, and the attachment type system on both splinted and separate implants [5,6,7,8,9,10,11,12].

Essentially, the mandibular overdenture serves as a complete denture with the special feature of gaining its biomechanical characteristics (support, stability, retention) from both soft tissue and implants [13]. As with every complete denture, a settling in period should be optional starting with its delivery. Such a period is believed to be between 2 to 3 weeks at the end of which the mandibular denture sits more closely to its supporting tissues than when initially delivered [14,15]. Such an intimate contact with the underlying tissue makes the load to be evenly dispersed throughout the denture intaglio surface. 

In implant-supported overdenture, a question arises regarding the load dissipation on both implants and tissue. In a situation where the attachment is connected simultaneously with denture delivery (immediate attachment incorporation), the stress distribution among the two elements (soft and rigid) may differ then in a case where a settling in period separates denture delivery and attachments incorporation (delayed attachment incorporation). When applying the settling in period (2–3 weeks) the unconnected overdenture is left to maximally use stability and support derived from the available soft tissues [16]. After completion of the settling in period, the well stabilized and fully supported overdenture will become activated through connection to the retentive attachment elements and its biomechanical features (support, retention, stability) may become fully reflected than with immediate incorporation, alongside denture insertion. 

A previously reported study showed that when activating a mandibular overdenture, the direct (chairside) attachment incorporation technique yielded less aftercare requirements compared to the indirect (laboratory) method [7].

To guarantee the long-term prognosis of prosthetic rehabilitation, especially implant supported overdentures (IOD), it is imperative to control implants and the residual ridge oral mucosa loading due to the unique complex implant–bone connection mechanism [11,12]. Specifically, residual alveolar bone quantity and quality, implants’ position, attachments type, and denture superstructure design. IOD treatment, as an implant tissue supported device, must be performed so that the implants and residual ridge are not subjected to excessive loads [4]. IOD can use a diversity of attachments (locators, balls, magnets, and bars). Most research on IOD attachments focused on the mechanical forces applied to the complex bone-implants [9,10,11,12]. Until now, to the best of our knowledge, no evidence exists regarding the timing impact of attachment system incorporation to a mandibular overdenture. This is a medical and prosthetic issue that seems to be critical for patient comfortable, function, and quality of life. The purpose of the present retrospective cohort study was to compare prosthetic aftercare between MISOD patients with simultaneous vs. three-week settling in period, prior to attachment incorporation.

## 2. Materials and Methods

### 2.1. Study Cohort

All individuals were included after evaluation of their medical histories, intra-oral dental examinations, and imaging (panoramic, orthoradial periapical radiographs and dental computerized tomography (CT) scans). 

All procedures were fully explained to the patients and the Ethics Committee of the Tel Aviv University approved the study (no. 7102008). All patients were treated by experienced oral-maxillofacial surgeons/periodontists and experienced prosthodontists/residents.

Inclusion criteria

Heathy individuals or with mild controlled systemic disease.Edentulous period of at least 12 months.Consecutive patients.Addressed the Oral Rehabilitation Department, School of Dentistry, Tel-Aviv University, Israel.Reduced stability of mandibular denture.Insufficient retention of mandibular denture.Classes III–VI resorption of the mandible according to Cawood and Howell [17].Presence of keratinized mucosa at the future implantation sites.

Exclusion criteria

Individuals with severe systemic uncontrolled disease.Stable mandibular denture without patient complaint.Lack of available data.Edentulous period < 12 months.Classes I-II resorption of the mandible according to Cawood and Howell [17].Absence of keratinized mucosa at the future implantation sites.

### 2.2. Surgical Treatment

Two implants per patient were inserted under local an aesthesia into the interforaminal region. All implants were placed as a submerged procedure. A total of 90 (44 group A vs. 46 group B) SEVEN ™ implants (MIS, Implant Technologies, Bar Lev Industry, Israel) were inserted. Implant lengths used were 10, 11.5, or 13 mm, depending on the residual alveolar ridge height of the mandible. Standard post-operative treatment composed of chlorhexidine 0.2% mouth rinses, analgesics, and antibiotics. All patients wore their old lower dentures (over the operated area) after the first stage of implant insertion.

Postoperative panoramic and orthoradial periapical radiographs were taken prior to implant uncovering. Three months after implant placement, second-stage surgery was performed. After 4 weeks, standard prosthetic treatment was carried out, being a new mandibular overdenture supported by ball attachments corresponding the implant system (Figure 1). 

### 2.3. Prosthetic Treatment

Treatment alternative was arbitrarily chosen by the clinician’s preference, resulting in 22 patients (group A) being treated with attachment incorporation simultaneously with denture insertion and 23 patients (group B) being treated with delayed attachment incorporation 3 weeks after denture insertion. Hence, group B replaced old dentures for new dentures without ball attachments connection (attachments connection was carried out 7 weeks after implants uncovering) and group A replaced old dentures for new dentures with ball attachments connected 4 weeks after implants uncovering.

Attachment incorporation into the mandibular implant overdentures were performed using a direct (chair-side) intraorally technique for both MISOD groups after joining the silicone liner to the metal housing and connecting them to the ball attachment which were screwed into the implants (Figure 2). A special separation rubber was seated around the ball abutment to avoid acrylic resin penetration (Figure 3a). The denture was placed and evaluated for space between the housings and the denture, confirming at least 2 mm acrylic resin over each housing. An auto polymerizing acrylic resin was placed into the housing holes in the denture-base (Figure 3b). The denture was placed with firm finger pressure and the patient was asked to occlude until polymerization was completed. Then, the denture was removed and the excess acrylic resin was removed from the intaglio and the outer surfaces of the denture. The denture was polished and inserted. The patient was provided with instructions on the insertion, removal, and maintenance of the prosthesis (Figure 3c).

In all patients, a balanced occlusion and monoplane acrylic teeth were used. The same laboratory fabricated all the overdentures. Moreover, this laboratory was involved in dentures repairs, reline, and revisions during the aftercare period. 

### 2.4. Follow-Up

From the first day that the patients visited the clinic up to 10 years, all surgical or prosthetic therapeutic interventions were recorded. The clinical follow-up included clinical examination radiographs (panoramic and orthoradial periapical radiographs) verifying marginal bone loss every 6 months in the first year and then every year (Figure 4). Prosthetic aftercare included routine recall visits every year. At a routine recall visit, implants’ stability and marginal bone loss, attachments, and the prostheses were checked. If needed, there were additional procedures for hygiene support and adjustment or repair of the mandibular overdenture.

### 2.5. Data Collection

All the data for the study were collected from the patients’ files. The recorded data for the present study included

Number of aftercare visits;Prosthetic dental treatment rendered:Pressure sores relieve;Liner changes due to loss of retention;Denture repair and reline.Gingival index measurements (0–3) according to Loe and Silness [18]. The gingival index was used for the assessment of prevalence and severity of gingivitis. Score 0 = Normal gingiva; Score 1 = Mild inflammation—slight change in color, slight edema, no bleeding on probing; Score 2 = Moderate inflammation—redness, edema, glazing, bleeding on probing; Score 3 = Severe inflammation—marked redness and edema, ulceration, tendency toward spontaneous bleeding.

### 2.6. Statistical Analysis

The Student’s t test and Fisher’s exact test were used for statistical analysis. The outcome variables were number of visits (pressure sore relief, liner changes, and denture repair) and gingival index. Independent variables included age, gender, and implant length.

## 3. Results

A total of 45 individuals (23 women, 22 men; mean age 65 ± 8 years, range 47–80 years) were included in the study (22—group A, simultaneous attachment incorporation, and 23—group B, delayed attachment incorporation). The demographic characteristics showed that there were no significant differences between both groups (mean age—63 vs. 65 years, *p* = 0.573; gender (male/female)—10/12 vs. 12/11). The mean follow-up of the entire study population was 84 ± 21 months, range 39–120 months. Statistical analysis revealed no statistically significant differences between the groups.

The mean number of visits dedicated to pressure sores relieve (3.63 ± 0.84 vs. 3.71 ± 0.61 for groups A and B, *p* = 0.581) (Figure 5). Pressure sores were found at the residual alveolar bone posterior to the implant location.

The mean number of visits dedicated to liner exchange due to loss of retention (2.09 ± 1.03 vs. 2.31 ± 1.04 for groups A and B, *p* = 0.487) deemed statistically insignificant (Figure 6).

The mean Gingival index also exhibited no statistically significant differences (1.3 ± 0.3 vs. 1.03 ± 0.2, *p* = 0.653) (Figure 7). 

No statistically significant differences between the groups were noted regarding denture repair and reline aftercare treatments (*p* = 0.318). The independent variables, including age, gender, and implant length, did not significantly affect the need for prosthetic aftercare.

## 4. Discussion

Most treatment failures are associated with mandibular dentures whose stabilization conditions are markedly poorer vs. maxillary dentures [19,20]. In order to prevent such undesirable results, aftercare measures are taken into account and deserve in depth research with all sorts of mandibular dentures. The present study revealed no significant changes in aftercare demands between the early and late incorporation groups. It appears that delivering the attachment system separately from the mandibular overdenture does not affect the occurrence of pressure sore spots underneath the dentures as well as matrix reactivation. No statistically significant differences between the groups were noted for the gingival index (*p* = 0.653). A low mean gingival score (1.3 ± 0.3 vs. 1.03 ± 0.2) was recorded for both groups. It can be speculated that the meticulous (every 6 months in the first year and then every year) maintenance (clinical follow-up) protocol was efficient. 

Post insertion pain evolves because of the nociceptive sensation beyond the pressure pain threshold (PPT) which triggers the edentulous patient to seek immediate remedy. It is noteworthy, however, that the presence of pain and the prevalence of mucosal injuries did not turn out to be significantly associated [19]. Pain may rise with or without mucosal ulceration. Settling of the denture base on its tissue foundation is important especially in cases with limited cushioning properties (resilience) in the supporting mucosa [21]. 

Stress concentration will lead to higher pressure loads and ultimately will surpass the PPT with evoked pain. An uneven stress distribution between the resilient component supporting the denture base and the non-resilient component will depend on the modulus of elasticity of both elements. Yet the literature reports the mucosal elastic modulus to range between values of 0.9 and 5.9 MPa through ultra—sound measurements [22], alongside mechanical recordings that confirm such magnitudes, namely 0.4–5.8 MPa [23], 0.4–2.7 MPa [24], 0.7–4.4 MPa [25], and 2.75–5 MPa [26]. In addition, the reports regarding the elastic modulus of the less rigid (yet more relevant) retentive element of the attachment system (Nylon rubber) reveal a value of 5 MPa [27]. Additional values in the literature report comparable modulus values with the ball attachment plastic resin matrix, namely 3.3 MPa [28] and the Locator attachment matrix Nylon resin—3 MPa [28]. Given the above values, the difference in the elastic moduli between the soft tissue support and the harder attachment system support is relatively negligible and indicates a uniform load distribution between them and between the loaded and unloaded implant overdenture state. These findings are in agreement with our results that showed no difference in timing of attachment incorporation into mandibular overdenture concerning aftercare prosthetic treatment. 

It is generally postulated that dentures attached to implants lead to more uniform distribution of loads to the mucosa as shown in a reliable histologic model in vivo [29]. Moreover, it is claimed that mandibular implant overdentures exert more consistent biting force vectors compared with conventional complete dentures, enabling less trauma to the denture bearing mucosa [30]. There should be no surprise that after completion of the overdenture activation in our study, no additional aftercare appointments took place, since the prostheses were well fitted with their tissue foundations.

Finally, a recent finite element model revealed that among the existing attachment systems, the ball attachment proved to be the most effective in reducing oral mucosa pressure compared to locator, magnet, and bar attachments [31]. These findings strengthen the results in our study, in which ball attachments were used in both experimental groups.

Numerous factors, including residual ridge morphology, implant position, retentive force, and maintenance capability, should be considered for the attachment selection. Pressure on the residual ridge mucosa was examined in many studies, and the results suggested that ball attachments might be the first choice to reduce oral mucosa pressure value during mastication [31,32,33,34]. Usually, oral mucosa pressure value should not exceed 200 kPa with any type of attachment [32]. Jawbone resorption with 2-IOD established no harmful effects if the oral mucosa pressure value did not exceed 200 kPa, and no stud attachment had unfavorable effects on the tissues and bone surrounding implants [33,34]. Those findings indicated that all the values acquired in our study should be sensible and imitate the clinical condition. 

Limitations of the present study include: retrospective nature, performed in only one treatment center, one type of attachment (ball attachment), and sample size. Further studies are recommended to compare the preferred timing of different attachments incorporation in 2-implants MISOD.

## 5. Conclusions

It can be concluded that prosthetic aftercare in MISOD wearers is similar whether a simultaneous or a three-week settling in period for the attachments’ incorporation is applied. It may be speculated that the ball attachment system uniformly disperses loads evenly between the attachment system and the soft tissue denture foundation, making it reliable and favorable for the MISOD patient. It may be recommended to incorporate the attachment simultaneous with denture delivery in order to gain denture stability, support, and patient comfort as early as possible.

## Figures and Tables

**Figure 1 jcm-11-03524-f001:**
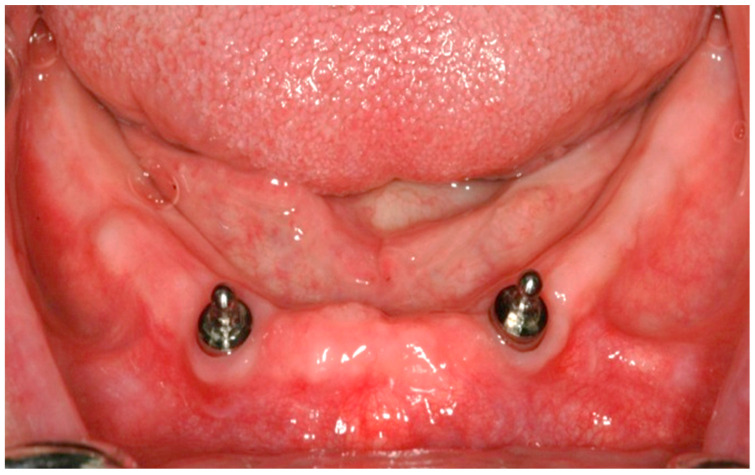
Two ball attachments connected to supporting implants.

**Figure 2 jcm-11-03524-f002:**
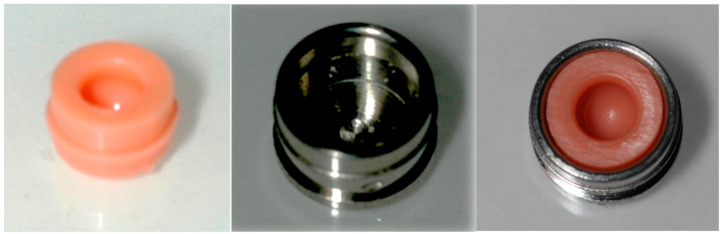
Silicone liner and metal housing.

**Figure 3 jcm-11-03524-f003:**
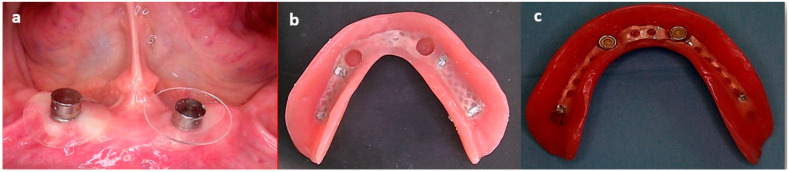
(**a**). Intraoral components before direct (chairside) attachment incorporation. (**b**). Denture base before direct (chairside) attachment incorporation. (**c**). Denture base after direct (chairside) attachment incorporation.

**Figure 4 jcm-11-03524-f004:**
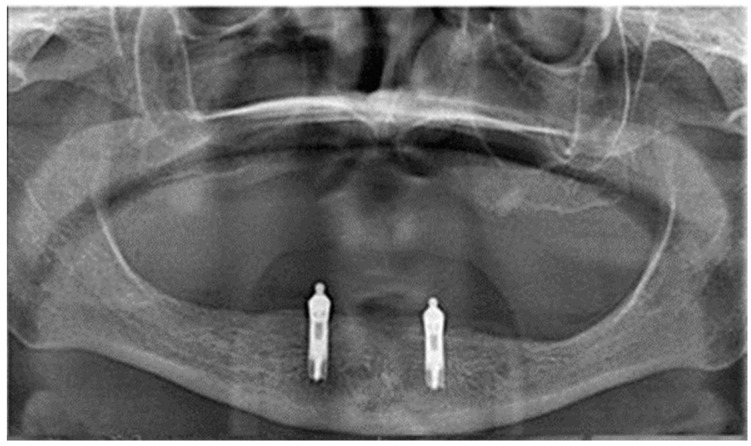
Panoramic X-ray. Post-operative, 60 months’ follow-up.

**Figure 5 jcm-11-03524-f005:**
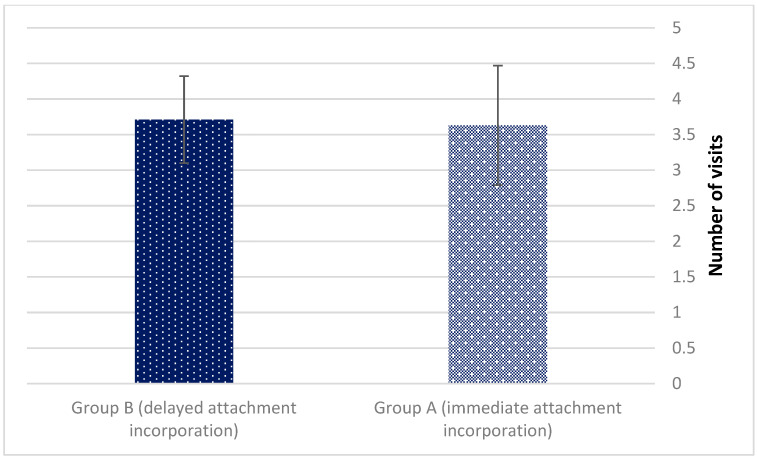
The mean number of visits dedicated to pressure sores relieve (*p* = 0.581).

**Figure 6 jcm-11-03524-f006:**
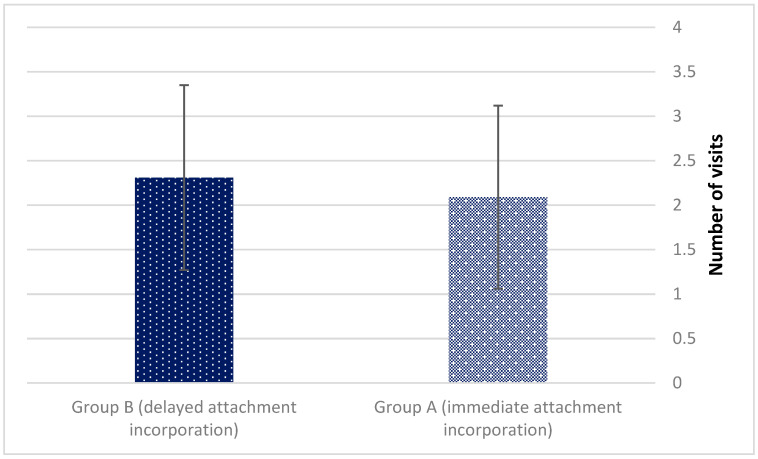
Liner exchange due to loss of retention *(p* = 0.487).

**Figure 7 jcm-11-03524-f007:**
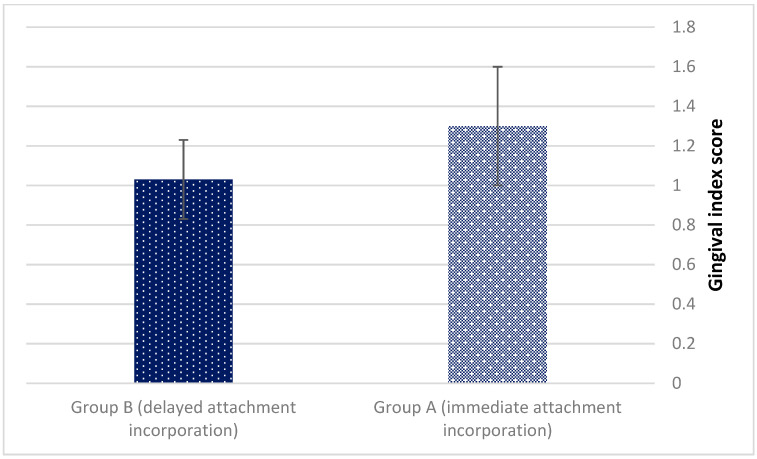
The mean Gingival index (*p* = 0.653).

## Data Availability

Data supporting reported results can be found in the table/graph.

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
