# Peer review of "Immediate versus Delayed Attachment Incorporation Impact on Prosthetic Aftercare among Mandibular Implant—Supported Overdenture Wearers"

_jcm, 2022, doi:10.3390/jcm11123524_

Round 1

Reviewer 1 Report

Dear authors, 

The study entitled " Immediate versus delayed attachment incorporation and its effect on prosthetic aftercare among edentulous mandibular implant-supported overdenture wearers" shows an interesting clinical investigation about the follow-up of patients using overdentures with ball attachment in two different protocols de confection. The study shows a good number of patients and evaluates important concepts. However, the authors should provide more information about the follow-up of these patients to have enough information that proves the non-significance between the two protocols proposed. Also, the authors showed only two figures showing the abutments applied, but no figures about the prosthetic rehabilitation, follow-up controls or final results between the two protocols. 

1- The authors should provide more pictures showing all the steps from this treatment and how the authors performed the follow-ups.

2- The figures are not mentioned in the text. It is difficult to follow where it should show the information.

3- The "mean gingival index" should be described briefly in order to simplify the understanding of the readers. 

4- The authors should show some extra information about the follow-up. Only the gingival index and number of visits did not show intense results to prove that the treatments are totally equal. The authors did not verify any radiographic parameters? 

The authors did not verify any issues in all the 45 patients over 10 years, such as peri-implantite or the necessity to change abutments?

5- In the discussion section, the authors should explore a little bit more in-deep that the protocols applied in this study were applied only with the abutment "ball attachment". Different results may be shown if the same study compares different abutments such as "locator", zirconia abutments or magnet. 

Author Response

Dear reviewer  thanks for the constructive comments ,answer were presented:

  • The authors should provide more pictures showing all the steps from this treatment and how the authors performed the follow-ups.
  • Pictures and text were added showing treatment steps

  • The figures are not mentioned in the text. It is difficult to follow where it should show the information.

2-  done

  • The "mean gingival index" should be described briefly in order to simplify the understanding of the readers. 

      3- Text was added as suggested “The gingival index was used for the assessment of prevalence and severity of gingivitis. Score 0 = Normal gingiva; Score 1 = Mild inflammation—slight change in color, slight edema, no bleeding on probing; Score 2 = Moderate inflammation—redness, edema, glazing, bleeding on probing; Score 3 = Severe inflammation—marked redness and edema, ulceration, tendency toward spontaneous bleeding.”

4- The authors should show some extra information about the follow-up. Only the gingival index and number of visits did not show intense results to prove that the treatments are totally equal. The authors did not verify any radiographic parameters? The authors did not verify any issues in all the 45 patients over 10 years, such as peri-implantite or the necessity to change abutments?

4- Text was added as suggested “. The clinical follow-up included clinical examination radiographs (panoramic and orthoradial periapical radiographs) verifying marginal bone loss every 6 months in the first year and then every year. Prosthetic aftercare included routine recall visits every year. At a routine recall visit, implants stability and marginal bone loss , attachments, and the prostheses were checked…….”

5- In the discussion section, the authors should explore a little bit more in-deep that the protocols applied in this study were applied only with the abutment "ball attachment". Different results may be shown if the same study compares different abutments such as "locator", zirconia abutments or magnet. 

5- Text and references (33-35)  were added “Numerous factors including residual ridge morphology, implants position, retentive force, maintenance capability, should be considered for the attachment selection. Pressure on the residual ridge mucosa was examined in many studies, and the results suggested that ball attachments might be the first choice to reduce oral mucosa pressure value during mastication.[32-35]. Usually, oral mucosa pressure value should not exceed 200 kPa with any type of attachment[33]. Jawbone resorption with 2-IOD established no harmful effects if the oral mucosa pressure value did not exceed 200 kPa ,also no stud attachment had unfavorable effects on the tissues and bone surrounding implants [34,35]. Those findings indicated that all the values acquired in our study should be sensible and imitate the clinical condition.

Limitations of the present study include: retrospective nature, performed in only one treatment center , one type of attachment (ball attachment) and sample size. Further studies are recommended to compare the preferred timing of different attachments  incororation in 2-implants MISOD.

Kawano, F. A study on pressures of supporting tissues under complete denture during functions-concerning the effects of the arrangement of artificial posterior teeth. J Jpn Prosthodont Soc 1987,31,726–39.

 Geckili, O.;Mumcu,E.;Bilhan,H.The effectofmaximumbite force,implantnumber, and attachment type on marginal bone loss around implants supporting mandibular overdentures: a retrospective study. Clin Implant Dent 2012,91–97.

 Cehreli, M.C.; Karasoy, D.; Kökat, A.M.; Akça, K.;, Ecker,t S. A systematic review of marginal bone loss around implants retaining or supporting overdentures. Int J Oral Maxillofac Implants 2010,25,266–277.

Reviewer 2 Report

Review for Manuscript ID: jcm-1747234entitled "Immediate versus delayed attachment incorporation and its effect on prosthetic aftercare among edentulous mandibular implant – supported overdenture wearers”

The manuscript is of interest and has merit for publication. However, there are points that need to be corrected as follows: 

1- The title is not clear and very long, I suggest “Impact of immediate versus delayed attachment incorporation on prosthetic aftercare among edentulous mandibular implant with supported overdenture wearers”

2- Punctuation errors can be seen, for example, in the first line of the abstract. The author needs to check the whole manuscript for that. 

3- In a result of the abstract there is not much data, authors advised to do so. 

4- Keywords: what is “direct activation” refer to? Better to be removed. 

5- Introduction: very short and does not justify the study. Further data regarding the problems in aftercare need to be added. Additionally, the rationale of the study needs to be clearly based on the gap available in the literature and why this study is necessary has to be added as well. 

6- All figures are not cited within the text. 

7- add OPG figures for a couple of cases to be seen radiographically.

8- Result: the first line is a phrase? 

9- Titles for the X-axis of Figures 3,4 and 5 need to be added. Moreover, the p-value needs to be added as well.

10- lines 149 to 151: where are the data? Add them as a table. 

11- Why the authors did not measure the amount of bone loss? I think this is more relevant in terms of aftercare than the gingival index as the gingival index tells what happens before two weeks rather than the whole period? So, I strongly recommend adding the amount of bone loss if there are x-ray for these patients. This will add value to the whole study. 

12- line 160: why you think there is no difference, explain.

13- Again, the discussion, in general, is very short and does not explain and compare the data with the previous studies.

14- Figure of “Institutional Review Board Statement” not necessary. 

15- How does this paper differ from the paper of reference number 7 by the two authors of this paper? 

BW, 

Author Response

June 5, 2022

Dear reviewer,

Thx for the constructive comments. The comments were addressed as follows -

1- The title is not clear and very long, I suggest “Impact of immediate versus delayed attachment incorporation on prosthetic aftercare among edentulous mandibular implant with supported overdenture wearers”

1- Title was modified as suggested- “Immediate versus delayed attachment incorporation impact on prosthetic aftercare among mandibular implant – supported overdenture wearers”

2- Punctuation errors can be seen, for example, in the first line of the abstract. The author needs to check the whole manuscript for that. 

2- Text was changed accordingly.

3- In a result of the abstract there is not much data, authors advised to do so. 

3- Text was added as suggested “. The mean follow-up was 84±21 months, range 39-120 months. The mean number of visits for: pressure sores relieve (3.63±0.84 vs. 3.71±0.61, p=0.581), liner exchange due to loss of retention (2.09±1.03 vs. 2.31±1.04 p=0.487) and gingival index (1.3±0.3 vs. 1.03±0.2, p=0.653) exhibited no statistically significant differences between the tested groups. No statistically significant differences between the groups were also noted for the denture repair aftercare treatments (p=0.318) and the independent variables including age, gender, implant length.”

4- Keywords: what is “direct activation” refer to? Better to be removed.

4- Keywords was corrected as suggested

5- Introduction: very short and does not justify the study. Further data regarding the problems in aftercare need to be added. Additionally, the rationale of the study needs to be clearly based on the gap available in the literature and why this study is necessary has to be added as well.

5-  introduction was enlarged and the rationale of the study was clarified “To guarantee the long term prognosis of prosthetic rehabilitation especially Implant supported overdentures (IOD), it is imperative to control implants and the residual ridge oral mucosa loading due to the unique connection mechanism of the complex implant-bone [11,12]. Specifically, residual alveolar bone quantity and quality, implants position, attachments type and denture superstructure design. IOD treatment, as an implant tissue supported device, must be performed so that the implants and residual ridge are not subjected to excessive loads [4].Implant supported overdentures (IOD) can use a diversity of attachments ( locators, balls, magnets, and bars) usually most research on IOD attachments has been focused on the mechanical forces applied to the complex bone-implants [9-12]. Until now, to the best of our knowledge, no evidence exists regarding the timing impact of attachment system incorporation to a mandibular overdenture .Medical and prosthetic issue that seems to be critical for patient comfortable  ,function and quality of life .”

6- All figures are not cited within the text.

6- All figures were added to the text. 

7- Add OPG figures for a couple of cases to be seen radiographically.

7- Clinical and radiographic figures were added

8- Result: the first line is a phrase? 

8- The text was modified as follows “Forty-five individuals (23 women, 22 men; mean age 65±8 years, range 47–80 years) were included in the study (22 - group A, simultaneous attachment incorporation, and 23 - group B, delayed attachment incorporation.)”

9- Titles for the X-axis of Figures 3,4 and 5 need to be added. Moreover, the p-value needs to be added as well.

9- All suggested values were added.

10- Lines 149 to 151: where are the data? Add them as a table. 

10- Data was clarified in the text.

11- Why the authors did not measure the amount of bone loss? I think this is more relevant in terms of aftercare than the gingival index as the gingival index tells what happens before two weeks rather than the whole period? So, I strongly recommend adding the amount of bone loss if there are x-ray for these patients. This will add value to the whole study. 

11- The study reports prosthetic aftercare and not prognosis. Consequently, MBL is not included because it is not an aftercare outcome parameter.

12- Line 160: why you think there is no difference, explain.

12- The following text was added “No statistically significant differences between the groups were noted for the gingival index (p=0.653). A low mean gingival score (1.3±0.3 vs. 1.03±0.2) was recorded for both groups. It can be speculated that the meticulous (every 6 months in the first year and then every year) maintenance (clinical follow-up) protocol was efficient.”

13- Again, the discussion, in general, is very short and does not explain and compare the data with the previous studies.

13- Discussion was enlarged as well as all parts of the manuscript

14- Figure of “Institutional Review Board Statement” not necessary. 

14- Figure was removed

15- How does this paper differ from the paper of reference number 7 by the two authors of this paper? 

 15- The study in ref.7 compare the prosthetic aftercare of direct vs. indirect attachment incorporation techniques to mandibular implant-supported overdenture.

While the present study compare prosthetic aftercare between MISOD patients with simultaneous vs. three-week settling in period prior to attachment incorporation. All patients received ball attachments using the direct (chairside) incorporation method. According to the conclusions of ref.7 and to our clinical experience over the years

We hope the revised version is suitable for publication

Reviewer 3 Report

The authors of this study compared the prosthodontic aftercare following different timings of direct matrix pick-ups on ball attachments in mandibular two-implant overdenture. The attachments were incorporated either simultaneously with the denture insertion or after a three-week, so-called “settling-in period” of the new overdenture. The reviewer read for the first time that a settling-in period could have advantages compared with an immediate incorporation of attachments. The authors hypothesized, that the fitting surface of the denture is better supported and stabilized by the mucosa after that and thus, the attachment can better reflect its purpose, i.e. denture support and retention.  To my mind, this assumption is highly speculative and therefore, the missing differences between the groups in terms of aftercare effort or gingival health were not surprising. 

Other points arose while reading the manuscript.

1.      A randomization of groups in a retrospective study is rather unusual. How was the allocation performed?  Were the participants randomized according to a list in a 1:1 ratio or as it happens?

2.      What is the meaning of “liner change due to loss of retention”? How were the matrices activated, by exchange of a housing insert?

3.      The authors gave not an account of overdenture relinings despite a mean observation period of 84 months. Prosthodontic aftercare should always include relinings if the fit of the denture base is deteriorated.

Author Response

June 5, 2022

Dear reviewer,

Thx for the constructive comments. The comments were addressed as follows:

The authors of this study compared the prosthodontic aftercare following different timings of direct matrix pick-ups on ball attachments in mandibular two-implant overdenture. The attachments were incorporated either simultaneously with the denture insertion or after a three-week, so-called “settling-in period” of the new overdenture. The reviewer read for the first time that a settling-in period could have advantages compared with an immediate incorporation of attachments. The authors hypothesized, that the fitting surface of the denture is better supported and stabilized by the mucosa after that and thus, the attachment can better reflect its purpose, i.e. denture support and retention.  To my mind, this assumption is highly speculative and therefore, the missing differences between the groups in terms of aftercare effort or gingival health were not surprising.

- We agree with you but as mentioned at the end of the introduction section “ Until now, to the best of our knowledge, no evidence exists regarding the timing impact of attachment system incorporation to a mandibular overdenture .Medical and prosthetic issue that seems to be critical for patient comfortable, function and quality of life .”

Other points arose while reading the manuscript.

  1. A randomization of groups in a retrospective study is rather unusual. How was the allocation performed?  Were the participants randomized according to a list in a 1:1 ratio or as it happens?
  2. The text was modified as follows - “Treatment alternative was arbitrarily chosen by the clinician preference”
  3. What is the meaning of “liner change due to loss of retention”? How were the matrices activated, by exchange of a housing insert?

2- Liner change mean exchange of the silicone liner only without exchanging of the metal housing, figures were added to the manuscript in order to clarify

  1. The authors gave not an account of overdenture relining despite a mean observation period of 84 months. Prosthodontic aftercare should always include relining if the fit of the denture base is deteriorated.

3- Dentures repair and reline appear as one of the outcome variables in the study (pressure sores relieve, liner changes, denture repair and reline) at the M&M and at the results sections.

We hope the revised version is suitable for publication.

Reviewer 4 Report

Journal: JCM

Manuscript ID: jcm-1747234

Type of manuscript: Research

Title: Immediate versus delayed attachment incorporation and its effect on prosthetic aftercare among edentulous mandibular implant – supported overdenture wearers.

This retrospective study was intended to compare the results of immediate versus delayed ball attachment incorporation in edentulous mandibular implant – supported overdentures.

This interesting and reliable study, based on long-term observations, evaluated a very interesting and pertinent topic in dental implant prosthetics. The conclusion of the study is that prosthetic treatment results of simultaneous at overdenture delivery vs. delayed attachment incorporation three-weeks after overdenture delivery are similar.

Only few remarks:

-        The abstract should contain the information about implant insertion protocol like “two stage protocol and three months after implant placement, the second-stage surgery was performed, etc. “ The problem is that now, it could be assumed that the article is about simultaneous implant loading with overdentures vs. delayed implant loading. This should be clarified.

-        In the methods section: it should be clarified whether all of the patients have been wearing lower dentures (over the operated area) after the first stage of implant treatment -  implant insertion or not having lower dentures during that time at all. Now it can be assumed (because is not clear) that the second group of the study replaced old dentures for new dentures without ball attachments and the first group replaced old dentures for new dentures with ball attachments after the second stage (with uncovered implants). Hence both of the study groups had old dentures replaced for new overdentures 4 weeks after the second stage of implant treatment.

-        The sentence in lines 93-93: “After 4 weeks, standard prosthetic treatment was carried out, being a new mandibular overdenture supported by ball attachments corresponding the implant system.” can be understood as - the second study group had ball attachments insertion 7 weeks after implant uncover while the first group had ball attachments insertion 4 weeks after implant uncover. It definitely should be clarified.

Author Response

Dear reviewer thanks for your constructive remarks:

-        The abstract should contain the information about implant insertion protocol like “two stage protocol and three months after implant placement, the second-stage surgery was performed, etc. “ The problem is that now, it could be assumed that the article is about simultaneous implant loading with overdentures vs. delayed implant loading. This should be clarified.

-     Abstract was corrected as suggested “Forty-five patients enrolled in this study , two stage implant insertion protocol was used while second-stage surgery was performed after three months.”

       More information about the surgical phase appear in details  at the M&M section :

” Two implants per patient were inserted under local an aesthesia into the interforaminal region. All implants were placed as a submerged procedure. A total of 90 (44 group A vs. 46 group B) SEVEN ™ implants (MIS, Implant Technologies, Bar Lev Industry, Israel) were inserted. Implant lengths used were 10, 11.5 or 13 mm, depending on the residual alveolar ridge height of the mandible. Standard post-operative treatment composed of chlorhexidine 0.2% mouth rinses, analgesics, and antibiotics. Postoperative panoramic and orthoradial periapical radiographs were taken prior to implant uncovering. Three months after implant placement, second-stage surgery was performed. After 4 weeks, standard prosthetic treatment was carried out, being a new mandibular overdenture supported by ball attachments corresponding the implant system”

-        In the methods section: it should be clarified whether all of the patients have been wearing lower dentures (over the operated area) after the first stage of implant treatment -  implant insertion or not having lower dentures during that time at all. Now it can be assumed (because is not clear) that the second group of the study replaced old dentures for new dentures without ball attachments and the first group replaced old dentures for new dentures with ball attachments after the second stage (with uncovered implants). Hence both of the study groups had old dentures replaced for new overdentures 4 weeks after the second stage of implant treatment.

-     Text was clarified as suggested “. Hence the second group replaced old dentures for new dentures without ball attachments connection and the first group replaced old dentures for new dentures with ball attachments connected .”

-        The sentence in lines 93-93: “After 4 weeks, standard prosthetic treatment was carried out, being a new mandibular overdenture supported by ball attachments corresponding the implant system.” can be understood as - the second study group had ball attachments insertion 7 weeks after implant uncover while the first group had ball attachments insertion 4 weeks after implant uncover. It definitely should be clarified.

- text was clarified as suggested “Hence the second group replaced old dentures for new dentures without ball attachments connection (attachments connection was carried out 7 weeks after implants uncovering ) and the first group replaced old dentures for new dentures with ball attachments connected 4 weeks after implants uncovering .”

Reviewer 5 Report

The manuscript contains numerous grammar and spelling mistakes.

Introduction - the authors should describe the clincal protocol for immediate/delayed attachment incorporation.

Material and Method - is there a registration number from the Ethics Committee?

Which were the exclusion criterias?

When were the prosthesis manufactured and inserted?

Was there a stratification considering associated diseases and medication?

Where were the pressure regions located?

Why were there no periimplantitis assessment?

Author Response

June 5, 2022

Dear reviewer,

Thanks for your constructive remarks. The comments were addressed as follows:

  1. The manuscript contains numerous grammar and spelling mistakes.

- Grammar and spelling mistakes were corrected.

  1. Introduction - the authors should describe the clinical protocol for immediate/delayed attachment incorporation.

- Text was clarified “In implant-supported overdenture a question arises regarding the load dissipation on both implants and tissue. In a situation where the attachment is connected simultaneously with denture delivery (immediate attachment incorporation), the stress distribution among the two elements (soft and rigid) may differ then in a case where a settling period separates denture delivery and attachments incorporation (delayed attachment incorporation). When applying the settling in period (2-3 weeks) the unconnected over-denture is left to maximally use stability and support derived from the available soft tissues [16]. After completion of the settling in period, the well stabilized and fully supported overdenture will become activated through connection to the retentive attachment elements and its biomechanical features (support, retention, stability) may be-come fully reflected than with immediate incorporation, alongside denture insertion.”

  1. Material and Method - is there a registration number from the Ethics Committee?

- Registration number was added – “7102008”

  1. Which were the exclusion criterias?

- Exclusion criteria were added

 “ Exclusion criteria

  1. Individuals with severe systemic uncontrolled disease
  2. Stable mandibular denture without patient complaint
  3. Lack of available data
  4. Edentulous period < 12 months
  5. Classes I-II resorption of the mandible according to Cawood & Howell [17].
  6. Absence of keratinized mucosa at the future implantation sites
  7. When were the prosthesis manufactured and inserted?

- Text was clarified at the M&M section “Treatment …………… resulting in 22 patients (group A) to be treated with attachment incorporation simultaneously with denture insertion and 23 patients (group B) to be treated with delayed attachment incorporation 3 weeks after denture insertion. Hence the second group replaced old dentures for new dentures without ball attachments connection (attachments connection was carried out 7 weeks after implants uncovering ) and the first group replaced old dentures for new dentures with ball attachments connected 4 weeks after implants uncovering .”

  1. Was there a stratification considering associated diseases and medication?

- The following text was added to inclusion criteria – “Heathy individuals or with mild controlled systemic disease

- The following text was added to  Exclusion criteria - Individuals with severe systemic uncontrolled disease

  1. Where were the pressure regions located?

- Text was added to the result section “Pressure sores were found at the residual alveolar bone posterior to the implants location.”

  1. Why were there no periimplantitis assessment?

- Text was clarify at the discussion section “No statistically significant differences between the groups were noted for the gingival index (p=0.653). A low mean gingival score (1.3±0.3 vs. 1.03±0.2) was recorded for both groups. It can be speculated that the meticulous (every 6 months in the first year and then every year) maintenance (clinical follow-up) protocol was efficient.”

More over the study is dealing with prosthetic aftercare. Peri-implantitis is a prognostic parameter.

We hope the revised version is suitable for publication

Round 2

Reviewer 1 Report

Dear authors, 

The current manuscript includes new pictures, discussion and methodological data and it sounds scientifically improved. 

I continue to think that the authors should find a better way to show your clinical pictures. The authors can create one figure with 3 or 4 images and improve the quality of the presentation of this study. Moreover, pictures of the techniques applied to control the follow-ups can be created using the same way (3 or 4 images in the same picture). However, after this correction, the authors included important data and content.

Author Response

One figure with 3 (Figure 3) was created for the presentation of this study as suggested.

Reviewer 2 Report

NIL

Author Response

Thank you very much

Reviewer 3 Report

The manuscript has been improved .

Author Response

Thank you very much

Reviewer 5 Report

Dear authors,

I read the revised version of your manuscript. 

Indeed, you answered to my questions, but I still stick to my first opinion, that your study should be extended in its number of subjects, protocol, and in that way the results will be consistent. 

Author Response

We feel that a cohort study containing 45 patients and 90 implants with a mean follow up of 7 years may contribute to the existing literature lacking studies on this topic